# Identifying the needs of people with long COVID: a qualitative study in the UK

Amy Miller ![ORCID],[1] Ning Song,[2] Manoj Sivan ![ORCID],[3] Rumana Chowdhury,[4] Melanie Rose Burke[1]

[1]School of Psychology, University of Leeds, Leeds, UK
[2]Leeds Institute of Health Sciences, University of Leeds, Leeds, UK
[3]Leeds Institute of Rheumatic and Musculoskeletal Medicine, University of Leeds, Leeds, UK
[4]Leeds Teaching Hospitals NHS Trust, Leeds, UK

**Correspondence to**
Amy Miller;
ps16am@leeds.ac.uk

## ABSTRACT

**Objectives** To identify the needs of people with long COVID (LC) in the UK.

**Design** Qualitative study using the Framework Analysis to analyse focus group discussions.

**Participants** 25 adults with LC aged 19–76 years including 17 men and 8 women. Average disease duration was 80.1 weeks.

**Setting** Eight focus groups were conducted in April 2023 online and in-person at the University of Leeds (UoL), UK. Recruitment routes included advertisement via Leeds Community Healthcare services, the English National Opera Breathe Programme and within the UoL.

**Results** Three key themes/needs were identified. (Theme 1) Support systems including community groups, disability benefits, clinical services and employment support should be accessible and tailored to the needs of people with LC. (Theme 2) Research should investigate the physiology of symptoms, new clinical tests and treatment interventions to improve clinical understanding of the condition and symptom management. (Theme 3) Societal awareness should be promoted via local and national initiatives to educate the public about the condition and reduce stigma.

**Conclusions** Participants experienced varied and individual challenges to daily life due to LC. There is a need for government acknowledgement of LC as a disability to ensure people with LC have access to disability support and legal protection. Policy development should be patient-driven and acknowledge the individual needs of people with LC in order to improve their quality of life.

## INTRODUCTION

Post-acute COVID-19 syndrome, commonly known as long COVID (LC), is defined as the continuation or development of new symptoms 3 months after the initial SARS-CoV-2 infection, with symptoms lasting for at least 2 months.[1] LC is a multifaceted syndrome with persistent, fluctuating and relapsing symptoms which commonly include fatigue, dyspnoea (shortness of breath), muscle and joint pain, difficulties in concentration and memory, headaches, heart palpitations, changes in mood and depression.[2] In 2023, the Office for National Statistics estimated that 1.9 million people living in private

## STRENGTHS AND LIMITATIONS OF THIS STUDY

⇒ This study followed rigorous methods to explore the experiences of long COVID.
⇒ The focus groups were accessible via in-person and online attendance, facilitating those with more severe symptoms to participate remotely.
⇒ A limitation of the study is the biased sample which consisted of mostly female participants aged 45–65 years old.
⇒ This study did not consider how additional long-term health conditions may have impacted participant's perceived needs.

households in the UK were experiencing self-reported LC.[3]

While the immediate focus of LC research has been on reporting the symptoms, prevalence and risk factors, a few qualitative studies have reported the lived experiences of people with LC.[4] The results reveal a debilitating condition which severely disrupts daily life due to the episodic and turbulent nature of symptoms, post-exertional symptom exacerbation and reduced physical and functional abilities.[5 6] Many people with LC experience stigmatisation in society, apathy and a lack of understanding from friends and family due to the 'invisible' nature of the condition, with negative impacts on relationships and well-being.[4 5] Furthermore, the impact on self-identity is commonly reported, whereby people with LC describe the difficulty of coming to terms with their new identity as an 'ill person' or their loss of professional identity where the condition impacted their employment.[4 5] The lack of access to services and limited treatment options contributes to feelings of hopelessness for recovery, poor mental health and well-being.[5] Such qualitative studies have revealed the challenging experiences faced by people with LC in accessing UK healthcare services in the first 6 months of the COVID-19 outbreak, including difficulties achieving a diagnosis, navigating and accessing services and being taken seriously by health professionals.[7 8] In the years

during the COVID-19 pandemic, 2020–2022, the UK government invested over £220 million in LC action plans designed to improve access to specialist clinics across the country, provide an online recovery programme and train general practitioner (GP) teams.[9] However, the outcomes of these developments on patient satisfaction are not reported, and there continues to be no approved treatment for LC.[10]

Three years following the first cases of LC,[11] reports continue to show lower life satisfaction, reduced happiness and greater symptoms of anxiety and depression in those with LC than those who have never been infected with the SARS-CoV-2 virus.[3] Despite years of research, it remains unclear whether people with LC feel adequately supported with their condition in 2023. Previous qualitative studies exploring the lived experience of LC have primarily focused on the difficulties of accessing healthcare and suggested quality improvements for healthcare services.[4 7 8] Yet, these studies have also revealed the widespread issues faced by people with LC including the impact on employment, relationships, stigmatisation and mental health, without identifying actionable strategies to tackle these issues. Therefore, there is an urgent need to comprehensively examine the broad range of needs of people with LC to address these issues and specify strategies to support symptom management, recovery and quality of life. It has been advised that guidelines should be developed through a participative and open process which is informed by the perspectives and expertise of individuals with LC.[12] This patient-centred approach has previously been implemented to design patient-generated quality principles to inform the development of LC services.[7]

By using qualitative methods in this study, we aimed to explore the multifaceted experiences and perceived needs of individuals with LC from a patient-centred perspective. We hope this will inform healthcare professionals, policymakers, researchers and support networks in implementing effective strategies to enhance the well-being of those affected by this persistent condition. We aimed to conduct a qualitative exploration using the Framework Analysis[13] as this method is rigorous and systematic, providing a structured framework for analysing qualitative data while allowing for flexibility and reflexivity in the analysis process. The Framework Analysis was considered to be appropriate for this study as it facilitates the synthesis of commonly described themes or concepts within participant's accounts, rather than deeply analysing and interpreting narratives as with other qualitative methods such as Narrative Analysis or Interpretive Phenomenological Analysis. The research question was; what are the perceived needs of people with LC?

## METHODS
### Participants
Participants included 25 adults with LC. Participants were eligible if they had self-reported LC symptoms and lived in the UK. Participants were recruited via advertisement in the Leeds Community Healthcare LC services, the Chronic Pain and Fatigue Network at the University of Leeds (UoL), the English National Opera Breathe Programme and via social media in attempts to increase visibility to a range of networks. Participants were reimbursed with a £20 Love2shop voucher. One participant did not provide demographic information, however, their data was included in the analysis due to the rich and relevant responses they provided.

### Focus groups
Eight exploratory focus groups, each containing two to five participants, were conducted in person (50%) and online via Zoom in April 2023 at the School of Psychology, UoL, UK. Online discussions were held to enhance the accessibility of the study ensuring that those living outside the region or those with impaired mobility could attend. Participants were informed that the discussions would be recorded but that any information they gave was confidential and would be anonymised. Each focus group lasted approximately 2 hours and was audio-recorded. Facilitators informally followed guidance questions (see online supplemental information) intended to encourage discussion around their main concerns, symptoms and ideas for support and interventions. Discussions were self-guided by participants and facilitators were advised not to influence participant answers or direct the conversation. Guidance questions were asked as each discussion topic came to an end. This allowed for organic discussions to take place, led by the participants.

### Data management and analysis
Audio recordings were transcribed verbatim and anonymised. Confidentiality was ensured by removing personal identifiers from the data and by limiting access to only those involved in the data analysis. The data analysis followed the procedural steps for the Framework Analysis[13] in four stages: (1) data familiarisation was conducted by reading and rereading the transcripts and listening to audio recordings from which initial codes were identified; (2) a framework for coding the data was developed and agreed on by each author; (3) the data was indexed through manually coding according to the framework in NVivo (R V.14.23.1, QSR International). To increase the reliability and validity of the data analysis, 25% of the data (2/8 transcripts) were independently double-coded by two analysts (AM and MS) and coding was compared. Any discrepancies were solved by consensus and the framework was altered accordingly. AM then applied the framework to all remaining transcripts to code the data. 20% of the coded data was randomly checked by the second data analyst (MS) in order to ensure consistency in the coding and reduce possible bias. Themes were generated through mapping and interpretation by summarising common concepts within the coded data, in relation to the research question (4). Data saturation was determined when no new themes were emerging from the analysis. To ensure the credibility of the data and analysis, the results were validated by discussing the themes within the research team and by seeking feedback from the participants. The research team involved in this study

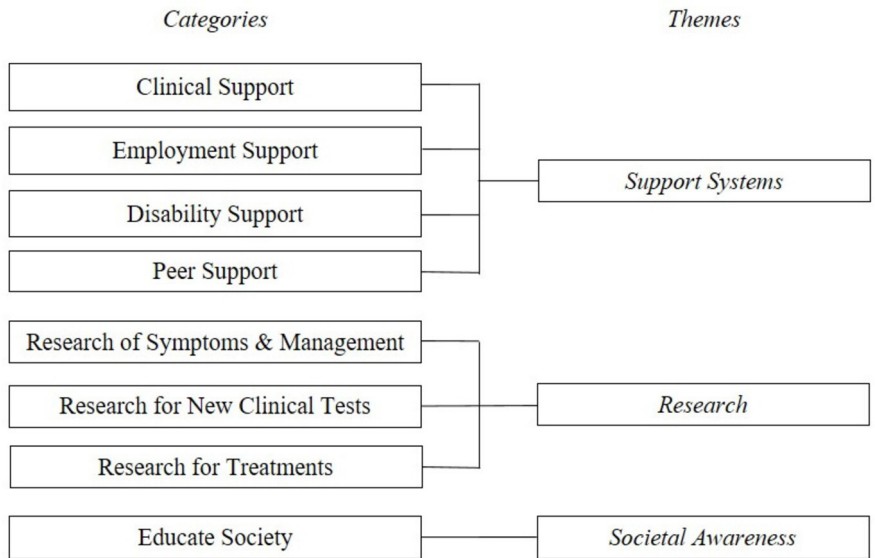

Categories Themes

**Figure 1** Summary of the categories included in each theme.

and manuscript refinement was interdisciplinary and interprofessional, combining expertise from LC rehabilitation, neurology, cognitive psychology and qualitative methods.

### Patient and public involvement statement

Participants were provided with a copy of the manuscript and were invited to review this prior to journal submission to ensure the findings appropriately represented their views. All feedback was positive and supported the findings.

### RESULTS

Participants were aged 19–76 years (M=43.6 years, SD=14.7; 17 women and 8 men). 22 participants (88%) had a clinical diagnosis of LC. The duration of symptoms experienced varied between 4 months and 2.8 years (M=77.96 weeks, SD=32.58). 21 participants had British nationality, 1 French, 1 Finnish and 1 Chinese.

The analysis produced three themes including needs for; (1) support systems, (2) research and (3) societal awareness, see figure 1. Please note, participant names are pseudonyms.

### Theme 1: Support systems

Participants expressed a need for greater support systems to facilitate access to healthcare, support daily functioning, maintain employment and receive the financial assistance. These concepts are explored within the categories; (1) clinical support, (2) employment support, (3) disability support and (4) peer support.

#### Clinical support

Participants described a long, challenging and uncertain route to accessing medical care with barriers at each stage of the process. This started with difficulty getting a GP appointment followed by scepticism and lack of empathy

from GPs when participants described their LC symptoms. Following GP referral, participants were left on a waiting list usually waiting months or up to a year until they could access a LC rehabilitation clinic. The waiting time served to exacerbate feelings of uncertainty about their route to recovery and participants felt 'left in the dark' about what they could be doing to self-manage or improve their symptoms in the meantime. As participants waited to be contacted, they built up inflated expectations of the potential that rehabilitation clinics could have on their health outcomes as the clinics were described as a 'secret club' which would 'hold the key' to their recovery. Once participants accessed healthcare services they described a lack continuity of care, as they were passed between medical specialists for examination. This intensified the burden participants experienced, as they felt responsible to manage their condition and route to recovery themselves alongside the challenges of the life-changing symptoms they experienced. Finally, participants described the inconsistency of care and investment offered across the country, referring to the healthcare offered to people with LC as a 'postcode lottery'. The negative experiences of waiting lists and lack of GP support were described by David and Ellie:

> [A GP told me] anyone who had anything in the last two years blames it on COVID… I'm not looking for a magic solution [from GPs]… Just information and maybe a little bit of reassurance. David

> There was a really big waiting, waiting list… it was just a complete void of what was gonna happen next… So just [needed] some kind of like, really basic information or updates on where you were in that process and what was gonna happen later. Ellie

Participants acknowledged the limitations of National Health Service funding and resources and practitioner's

limited medical knowledge of LC due to the novelty of the condition. However, they suggested various improvements for healthcare. First, GPs should have appropriate training of LC in order to provide more targeted advice, refer patients to the correct services and adopt a more empathetic approach. Another common suggestion was referral to an infectious disease specialist, who could follow their progress and provide a continuity of care, relieving individuals of the burden to manage their route through medical care alone. Physiotherapy and occupational therapy were suggested to provide training in breathing techniques, physical exercise and supporting individuals to complete activities of daily living. The need for psychological counselling was highlighted to support people with LC with symptoms related to mental health including depression and anxiety. In addition, psychological therapies should support the grieving process associated with losing the life individuals had before LC and accepting a new identity as a 'sick person'. Overall greater empathy, information availability and wide ranging therapies were required by participants to be supported by health services. These points are highlighted by Rachel and Angela:

> To have someone who is primarily looking after you and analysing everything and that kind of overview and then you can go to cardiology or to neurology, but then to kind of go back to somebody. That would be amazing. Rachel

> It was really difficult to accept my new identity as a sick person, and someone who was always letting people down… I felt kind of embarrassed almost about having long COVID… having counselling really helped me to work through that. Angela

### Employment support

For many participants, their ability to work was impacted by their LC symptoms, particularly breathlessness, fatigue, dizziness, heart palpitations and 'brain fog'/cognitive deficits. Some participants were unable to continue working or found difficulty concentrating with regular rest required throughout the working day. Long daily commutes were especially noted as a contributing factor towards chronic fatigue which subsequently impacted performance at work the following days. Participants found that colleagues often did not understand the condition leading to friction and hostility, or awkward moments in which they had to explain to others why could not attend in person meetings or events. Managers requested medical verification of participants' condition which they were unable to provide if their medical test results were 'normal', contributing to a sense that employers did not believe their condition. In some cases, participants filed issues against their employer, further exacerbating the stress experienced while managing their symptoms. The experiences of conflict at work associated with LC were described by Alison and Alice:

> But [medical tests] come back normal and you've got Long Covid. Okay, well, that doesn't really bode well for my boss who wants to see some kind of medical evidence that I've got Long Covid because he thinks I'm lying. Alison

> I was off work two years and I had to fight with work because they couldn't understand why I couldn't come back… they were quite happy for me to just disappear. But I took a grievance out and I've got a part time [job]… I am absolutely ecstatic because I really, really didn't think I would get back into the workplace. Alice

Oftentimes, the impact of the condition on participants' ability to work had challenging implications on individuals' self-image and sense of self-worth. Participants described the frustration of being unable to work at the level they could before LC, as they felt they were 'not achieving their full potential' and were subsequently 'failing'. There was also a sense of loss of identity, as many participants tied their professional identity with their personal identity and sense of self-worth which were negatively impacted following LC. Some participants described themselves as 'unreliable', reflecting on the shame they felt at 'letting others down' at work, as described by Thomas:

> I was one of the top salespeople in my particular department… And then when COVID hit, everything stops… I feel like I'm probably one of the most unreliable people there because no one ever knows when I'm going to be there. People at work just don't get it all. It kind of can hit quite hard. Thomas

Participants identified various strategies to support people with LC in the workplace. Mandatory training for employers in dealing with chronic health conditions was noted as a strategy to promote a more understanding and empathetic approach. People with LC should be supported to work through a phased return to work programme, with options for 'hybrid working', working from home and part-time work. Some noted small adjustments that would be beneficial, such as providing a quiet area for people with LC to work and opportunities for rest and breaks. Furthermore, access to a workplace occupational therapist should be available where possible. It was also reported that LC should be recognised as an industrial injury where the virus was contracted in the workplace. Margot identified that these changes required enforcement through employment law:

> You need to have a much more understanding, forgiving approach to it from employment law. And you know, the support you get in terms of paid sick leave or shorter hours or different ways of working. Margot

### Disability support

Participants described wide-ranging, debilitating and life-changing symptoms following LC which impacted

mobility, the ability to complete activities of daily living and employment. Many participants described working reduced hours or ending their employment following LC, resulting in a reduced income and financial shortfall:

I had to stop on my full time job and get a part time job… which was not convenient for my financial situation. So I really need more intervention in my financial level. Max

Therefore, there was a strong need expressed for government action to recognise LC as a disability in order to provide people with LC with disability support for the varied and personal challenges they faced. For example, financial support through the Personal Independence Payments (PIP) or subsidies:

The government also needs to recognise that long COVID, it is a serious illness and they need to sort that out. There needs to be some sort of a subsidy. Lisa

However, participants expressed confusion regarding whether LC was currently classed as a disability and a lack of information about the government support available to them. Specifically, there was some debate about whether the Blue Badge for accessible parking was available to people with LC, or whether this may be issued on the grounds of particular symptoms such as anxiety or immobility. Participants were reluctant to identify themselves as 'disabled', which appeared to reflect the difficulty they faced with their changing self-identity as a 'sick person'. Alison described her confusion about the government support available to her and why the Blue Badge would meet her needs:

I'm not actually sure whether it's classed as a disability… because I asked about getting a blue badge… I got told I don't think you'll get one because it's not been classed as disability yet. If they can just tell you whether it's classed as a disability… And there were some information out there about it. Alison

I wouldn't say I was particularly disabled. But sometimes I think something like that [blue badge] will be quite handy because I'm terrible with inclines. Alison

Household help including cleaning, grocery shopping, gardening and dog walking were listed as ideas for local community care to improve quality of life by reducing daily demands and leaving more time and energy to do what they enjoy:

If we could have like funding and help for a cleaner to come that would take pressure off me… and do things you enjoy more, like walking with friends. Rose

### Peer support

Many participants expressed the need for more LC support groups, both online and in-person. They highlighted the importance of building a network to connect with others who could understand them and create a sense of community. It was highlighted that this could help to alleviate the sense of loneliness and isolation experienced with LC, as others in the group could understand and empathise with the symptoms they were experiencing and the impact it had on their lives. Within a community support group, participants could share their knowledge of this poorly understood condition. This included exchanging ideas and tips for symptom self-management and opportunities to connect, receive support or participate in research. By connecting with others, participants felt that this 'strengthened their fight' against the condition and were more optimistic about their recovery. Kelly and Graham described the benefits of attending support groups:

It's so nice to speak to people that are experiencing something similar to you. And it makes you feel so much less isolated. Kelly

Even just knowledge sharing like we've been doing today, just tips for coping. Graham

However, it should be noted that one participant reported mixed feelings about peer support groups following an experience of meeting others with less severe symptoms than herself. This shows how if an individual does not connect to the group due differing symptom severity, this could exacerbate feelings of isolation in their condition, breaking down the sense of community an individual is seeking. This was articulated well by Kelly:

If somebody's not as bad as you, it… makes me feel a bit like, oh, why have I got it so bad? Is there something wrong with me? And that had quite a big impact on myself because I thought we were all in this together. Kelly

### Theme 2: Research

Participants expressed the needs for research to examine (1) symptoms and management, (2) new clinical tests and (3) treatments.

Participants described a lack of understanding of LC in the medical community which could be addressed by researching the condition. Specifically, participants wanted research to investigate the causes and physiology underlying the common symptoms such as fatigue, breathlessness, post-exertional malaise (PEM) and brain fog and the more unusual symptoms such as dermatological problems. It was expressed that a greater understanding of the basis of their symptoms would help participants cope with their condition and manage their symptoms:

the more they understand about the physical, especially the lungs, the more they understand that, the more they [medical professionals] can relay that to you, the better you're gonna be mentally knowing what's wrong, what, what you're up against. Peter

Some kind of understanding or research on the reasons for the kind of what I feel is like a physical collapse from being really quite fit to feel like there's

a disconnect between our different parts of the body that are needed for physical exercise. Samantha

## Research for new clinical tests

Many participants recalled the frustration and disappointment of receiving 'normal' results following medical tests, which did not explain the chronic symptoms they were experiencing. Consequently, participants wanted research to focus on developing new clinical tests that are better able to detect changes or damage in their body associated with their symptoms:

> I think as well, like, more tests, because every test that you do it's fine, like there's nothing… actually that they can say but then… it just seems like the wrong tests. Rachel.

## Research for treatments

By improving the understanding and detection LC-related changes to physiology, it was proposed that this could inform research into new treatment options. Drug treatments, cognitive training and physical exercise were suggested as possible interventions to trial, however, it was recognised that there may never be 'a cure' for the condition. Despite this, many participants expressed a strong willingness to participate in research and were keen be informed of the outcomes. They felt that participating in research would be a worthwhile and positive experience, as they could use their challenges to contribute towards a greater cause:

> I'm happy to give my time and my blood and you know to research, because if it helps us finding cures and solutions… Make something useful out of something, yeah, that's not great to experience. Rose

While participants were motivated to take part in research, they highlighted the importance of tailoring the study design to accommodate their needs. Researchers should recognise the additional energy required for people with LC to commute and take part in studies and possible accessibility requirements. Participants explained their ability to participate in research may depend on whether the study intervention fits into the current advice they are following and their work routines. Furthermore, researchers should be flexible in supporting those with LC to participate in studies due to the fluctuating symptoms, relapses and PEM which may affect their ability to take part:

> I guess just listen to your participants when they come for that day, treat them on an individual day-to-day basis because they might come one day and they're feeling really good and they might come the next day and feel like [rubbish]. Kelly

## Theme 3: Societal awareness

The need for greater societal awareness was expressed by participants to address the judgement they experienced from others about their condition. Suggested strategies to promote awareness and understanding of LC centred around various initiatives to 'educate society' as described below.

## Educate society

Participants experienced judgement and stigma of their condition from others, centred around a lack of understanding of LC in society. It was acknowledged that LC is an 'invisible' illness, in which the symptoms may not be immediately obvious to others. However, this required participants to repeatedly 'explain themselves' and their condition which they found frustrating and demeaning. Participants found that others were dismissive of their symptoms or did not believe the condition existed, leading to judgements that they were 'lazy' or 'overdramatic' if the condition affected their daily life or ability to work. Furthermore, a lack of understanding of LC caused some people to have fear that the condition was contagious. David and Alison the lack of understanding they experienced from others:

> A very significant number of people don't take it seriously. They think it's just being lazy or a bit shiftless, and you can't be bothered to do stuff. If we could kind of somehow get rid of that stigma… that people wouldn't willingly put themselves through these kinds of life limiting situations, I think that would be really helpful. David

> Sometimes you'll tell somebody you've got Long Covid and they jump back three paces because they're gonna catch it… there's a lot of ignorance around it. Alison

Therefore, participants expressed a need for greater awareness of LC in society by educating the general public about the severity of symptoms and prevalence of the condition in society. A number of suggestions were made for promoting awareness of LC, including publicising the experiences and common symptoms via local and national news reports and TV programmes. It was suggested that educating the public about LC may trigger action via government funding for research and support:

> Maybe it needs to be on the news… Maybe it needs to be shown how many people are actually suffering with it… and then people might be more understanding. Megan

> …get it out there, get it in the news, get it publicised and I'm sure eventually it would filter through and people will be aware of it and start, you know, start start doing things. Lisa

They reported that better public understanding of the condition would make others more sensitive and empathic to their condition and reduce the need to regularly explain the condition to others:

> So I think if you could make it more like serious, almost… other people would then take it more

seriously, and then that would then help us because we don't have to explain ourselves… General awareness, but also awareness how serious it is and how like it does affect us. Rachel

Art, poetry and blogs depicting the experiences of LC were also suggested as possible media for influencing an emotive response from the pubic and encouraging a more empathic approach:

I think maybe art can bridge some of that gap a little bit… I wrote some poetry about my experience with Long Covid and showed it to my friend. They said it helped them kind of empathise a little bit more… That's another way that our experiences can be communicated to the wider public. Angela

## DISCUSSION

The purpose of this qualitative study was to investigate the perceived needs of people with LC. The findings revealed an overarching message that people with LC require greater support and awareness of their condition. First, participants expressed a critical need for robust support systems, including accessible healthcare, tailored employment assistance, disability support and peer support through organised groups. Second, there was a strong call for research to explore the underlying mechanisms of LC symptoms, develop new treatments and improve clinical testing. Finally, participants emphasised the importance of raising awareness and promoting empathy for LC within society through public education initiatives. These findings underscore the complex challenges faced by individuals with LC and highlight the urgent need for targeted interventions, research initiatives and societal awareness efforts to address the diverse needs and enhance the quality of life of people with LC.

### Comparison of findings with existing research

There are a few qualitative studies that have explored the lived experience of LC with which we can compare our findings.[4 5 7 8] The findings of this study are consistent with previous qualitative research demonstrating the general lack of understanding of LC. This results in experiences of judgement and stigmatisation in society and a lack of belief or empathy from employers and GPs, further contributing to the sense of burden and isolation associated with LC.[4 5 7 8] Participants in this study offered many suggestions for educating society and promoting awareness of LC including news reports, TV programmes and art. The most-reaching approach to enhancing the credibility of the condition, identified by participants, was government recognition of LC as a disability. Furthermore, the barriers to accessing healthcare found in this study reflect those described in previous studies including long waiting periods for referral, a lack of continuity of care and the lack of understanding or treatment options for the condition.[7 8]

Many of the suggestions provided by participants in this study echo previously proposed quality principles for LC services.[7 14 15] Specifically, the need for greater access to healthcare, multidisciplinary rehabilitation services, clinical responsibility and further development of medical knowledge and therapies offered. Further to these points, we suggest improving access to information during referral periods such as signposting to online information and community support to ensure people with LC are supported during this time.

### Implications for policy change

The unique findings of this study are the numerous suggestions made by participants for policy changes to support people with LC. Participants expressed their needs for the Blue Badge for accessible parking, financial support, inclusive employment law and an overarching need for legal recognition of LC as a disability. The Equality Act 2010 defines a disability as 'a physical or mental impairment that has a substantial and long-term negative effect on the ability to do normal daily activities'.[16] In May 2022, the Equality and Human Rights Commission confirmed that LC does not automatically fall under the list of conditions that class an individual as disabled under the Equality Act 2010.[17] However, the Citizens Advice lists LC as condition which may cause impairments, possibly qualifying an individual for disability support and protection from discrimination.[18] Without formal acknowledgement of LC as a disability, people with LC face uncertainty and restrictions to accessing disability support and a greater risk of discrimination in the workplace and society.[19]

Here we highlight the vital importance of revising the current Equality Act 2010 to include the classification of LC as a disability. This formal acknowledgement of the condition is an actionable step for the government to provide credibility, legal protection and disability support to make a qualitative improvement to the quality of life of people with LC.[19] Under the Equality Act 2010, employers have a legal responsibility to make reasonable adjustments to support employees with disabilities to continue working. Therefore, legal recognition of LC would incentivise organisations to be flexible to meet the needs of people with LC[16] and follow guidance for supporting individuals to return to work, such as those published by The Society of Occupational Medicine.[20] Access to the Blue Badge for accessible parking would facilitate independence to access services, activities and opportunities to socialise outside the home. Financial assistance such as statutory sick pay or PIP should be accessible if the condition affects individuals ability to work. Wider effects of this policy change would hopefully include greater sympathy, concern and belief of the condition from family, friends and society and funding opportunities for community-based care, support groups and research. Ultimately, legal recognition of LC as a disability requires clear diagnostic criteria supported by clinical research into the underlying mechanisms of persistent symptoms.[19]

## Strengths and limitations

Strengths of this study include the accessibility of the focus groups via a choice of in-person or online attendance and the use of a second analyst to quality assess the framework and analysis to reduce bias in the coding procedure. The researchers and facilitators in this study had no personal experience of LC symptoms facilitating a more open-minded approach to the interpretation of participant's experiences. However, inevitably this meant that they lacked shared experiences which could add richness to the interpretation. The inclusion of multiple facilitators in discussion groups may have introduced different styles of questioning and focuses of conversation. Despite this, there were common patterns in the data that emerged across discussion groups. The current study could be limited by a biased sample which consisted of a 2:1 ratio of female to male participants and predominantly middle-aged participants (50%). LC is more prevalent in women than men[21] which may reflect a true gender difference or a reporting bias. Despite our efforts to advertise the study across the UK, most participants were located in West Yorkshire and therefore the results may not reflect those of people with LC across the UK. Furthermore, participants of this study were self-selecting therefore the sample may only reflect those who felt they had the mental capacity, time and opportunity to take part. This study did not consider how additional long-term health conditions may have contributed to participant's perceived needs, therefore it may not be possible to distinguish which needs are driven by LC or other comorbidities. This study did not explore the impact of participant's medical history, such as COVID-19 vaccination status, on the results and this could be considered as a potential limitation. However, the focus of the study was to explore commonly described experiences and needs in LC, rather than to analyse the impact of the diversity of individual factors or medical history on perceived needs.

## CONCLUSIONS

In light of our investigation into the perceived needs of individuals with LC, our findings highlight a critical imperative for policy reform to recognise LC as a disability. Our investigation has brought to the forefront the multifaceted challenges faced by people with LC, from navigating employment uncertainties to managing daily activities and social interactions. By recognising LC as a disability, policymakers can pave the way for the implementation of comprehensive support systems that are tailored to the needs of individuals living with LC. Central to this is the imperative for policy initiatives to adopt a patient-centred, collaborative approach that actively involves individuals with LC in the decision-making process. Only through such inclusive policies can the diverse and individual needs of people with LC be adequately addressed. By bridging our findings with actionable policy recommendations, we aim to initiate meaningful strides towards enhancing the quality of life of individuals living with LC.

This study recommends future studies should continue exploring these experiences in order to inform the development of innovative treatments and interventions to support the health and well-being of people with LC.

**Correction notice** This article has been corrected since it was published. Licence updated to CC BY on 2nd August 2024.

**Acknowledgements** We would like to thank all of the participants who took part in the focus groups, especially as many individuals were experiencing physical and cognitive challenges at the time of the study. We would like to thank the University of Leeds for funding this research. Thank you to Sabina Meehan for her assistance with conducting the quality assurance of the framework and coding. Thank you to the facilitators Megan Smith, Shihui Yu and Soumya Shetty for their assistance with data collection.

**Contributors** AM is responsible for the overall content as the guarantor. MRB, MS and RC gained funding for the study. AM and MRB conceived the idea and conducted the study. NS informed the methods for qualitative data analysis. AM conducted the transcription, data analysis and wrote the first draft of the manuscript. All authors contributed towards the refinement of the manuscript.

**Funding** This study was supported by the University of Leeds via the Wellcome Institutional Translational Partnership Award, grant number 219420/Z/19/Z. Funders had no role in the planning and execution of the study or writing up of the paper.

**Disclaimer** This article reflects the views of the authors only, not the institutions or funders.

**Competing interests** None declared.

**Patient and public involvement** Patients and/or the public were involved in the design, or conduct, or reporting, or dissemination plans of this research. Refer to the Methods section for further details.

**Patient consent for publication** Not applicable.

**Ethics approval** The study was approved by the University of Leeds Psychology Ethics Committee on 3 January 2023 (PSC710). Participants gave informed consent to participate in the study before taking part.

**Provenance and peer review** Not commissioned; externally peer reviewed.

**Data availability statement** Data are available upon reasonable request.

**ORCID iDs**
Amy Miller http://orcid.org/0000-0002-8549-5158
Manoj Sivan http://orcid.org/0000-0002-0334-2968

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
