## [Reviewer comments · BMJ Open]

ARTICLE DETAILS

TITLE (PROVISIONAL)	Identifying the Needs of People with Long Covid: A Qualitative Study in the UK
AUTHORS	Miller, Amy; Song, Ning; Chowdhury, Rumana; Burke, Melanie; Sivan, Manoj

VERSION 1 – REVIEW

REVIEWER	Matthew Bonn Canadian Association of People Who Use Drugs, Dartmouth, Nova Scotia
REVIEW RETURNED	29-Dec-2023

GENERAL COMMENTS	December 29th, 2023 Thank you to the editors at BMJ and the authors on the submitted manuscript for allowing me to participate in the peer review process for this manuscript. I feel like I have sufficient expertise to effectively and properly review this submission. My proposed decision is going to be accept this manuscript once minor revisions have been made. I'm looking forward to reading any revisions made by the authors. Thank you, Reviewer MINOR REVISIONS (MR) ABSTRACT MR #1: I don't think we should be adopting any new acronyms in the academic literature that start with people. It takes away from the person first language that the authors are trying to accomplish. I would suggest writing it out throughout the manuscript. MR #2: Did the authors collect demographic information on the participants besides gender? If so it should be in the participants section. INTRODUCTION MR #3: Line 43-54, where it says "participant-centred approach" this should be "patient centred approach" and there should be a couple examples of other types of illnesses that adopt this approach such as hepatitis c and opioid use disorder. https://www.cmaj.ca/content/191/17/E460.short
---

	METHODS MR #4: Page 6, line 14-18: Were any patients asked to co-author the manuscript? Given them the proposer recognition for their involvement in the submitted work. RESULTS MR #5: Page 9, line 5: Should this say “infectious disease” specialist, instead of “specialist” ...just saying specialist is extremely broad. MR #6: Page 9, line 35-45: Under the community support heading, would peer support be more appropriate? It seems they want to connect with other people with long COVID, that’s what really stands out to me. MR #7: There are a lot of oxford comma’s missing through the manuscript, I think the authors should carefully review it and update this oversight. Major revisions that I should be considered: Major Revision #1: There is no mention of vaccines throughout the manuscript...do we know if participants have been vaccinated or were asked any questions about vaccinations? I think there should be some mention of vaccinations in the introduction and discussion section. If there weren’t any questions asked to the participants, then maybe the authors could explain their rational and include it in the limitation section. Major Revision #2: There is no mention of any ethics board approval for this study...is this an oversight or was there no obtained?
--	---

REVIEWER	Shi Yin Chee Taylor’s University
REVIEW RETURNED	02-Jan-2024

GENERAL COMMENTS	Identifying the Needs of People with Long Covid: A Qualitative Study in the UK Manuscript No: bmjopen-2023-082728 Thank you for the opportunity to review this manuscript. While there are commendable attempts to cover a much-needed area in this field, I believe that major revisions are necessary to meet the publication standards to improve the manuscript’s clarity, depth, and comprehensiveness. While it offers a meaningful contribution to understanding Long Covid, it could benefit from improvements in several areas, including a deeper exploration of themes, more rigorous methodological detailing, and a clearer articulation of conclusions and implications. Addressing issues related to participant diversity, grammatical accuracy, and consistent formatting will further strengthen the manuscript's impact and readability. Please see my comments below:  1. The title could be more specific about the study's scope or the aspects of Long Covid being addressed. The author/s could refine the title to specify the key focus of the study, such as "Exploring Support and Healthcare Needs of People with Long Covid in the UK." 2. The study’s exclusion of the impact of additional long-term
--

	health conditions on participants' needs might limit the comprehensiveness of the findings. This omission could be critical, especially given the multifaceted nature of Long Covid and its intersection with other health issues. 3. On page 4, not specifying what "UK" stands for in the manuscript could be considered an oversight, particularly for an international audience. While "UK" commonly refers to the United Kingdom, it is a best practice in academic writing to avoid assumptions about readers' familiarity with abbreviations. The author/s should initially define this abbreviation, like stating "United Kingdom (UK)," before using the abbreviated form throughout the manuscript. 4. The introduction may lack clarity or focus on key research gaps in Long Covid research. The introduction broadly discusses Long Covid impacts without honing in on specific gaps the study aims to address. 5. There is limited context on the current state of Long Covid research and patient experiences provided by the author/s. General statements about Long Covid without detailed background on existing research or patient experiences. 6. The justification for the study by the author/s may not be clearly articulated. The rationale for conducting this study is not explicitly stated in the introduction. 7. Possible misalignment between study objectives and introduction content. The introduction does not directly lead to the specific objectives of the study. The author/s need to ensure that the introduction seamlessly leads into the specific objectives of the study, making a direct connection between the background provided and the study's goals. 8. The choice of qualitative method is stated without detailed justification. Provide a rationale for why the qualitative method was chosen over other potential methods. 9. Limited details on the criteria for participant selection. "Participants were eligible if they had self-reported LC symptoms and lived in the UK". 10. Further elaborate on how participants were selected, including any exclusion criteria. "Eight exploratory focus groups... were conducted in person and online". Provide more detailed descriptions of the data collection process, including how discussions were guided. 11. Discuss ethical considerations in more depth, including how participant confidentiality and consent were handled. 12. Unclear representation of diverse experiences in participant selection. Please clearly describe efforts to include diverse demographics and acknowledge any limitations in representation. 13. Vague explanation of data analysis steps: "The analysis followed the procedural steps for the Framework Method". It would be great if the author/s could provide a clearer trail of each step of the data analysis process, especially leading to how themes were derived. What about data saturation? Kindly elaborate how and when data saturation was reached and how this was determined during the analysis. 14. Measures to ensure reliability and validity are not explicitly discussed. Discuss the steps taken to ensure the reliability and validity of the data collected and analyzed. 15. The results chapter may lack specific details or examples with insufficient use of direct quotes from participants to support the themes. Vague descriptions of participant experiences. Themes are stated with minimal direct quotations to substantiate them. The author/s must integrate more direct quotes from participants to
--	---

	illustrate points, provide authenticity and depth to the themes. 16. Superficial theme development and analysis of participant responses. The author/s need to provide deeper analysis and critical insights into the data. For instance, the manuscript states themes such as "Support Systems" and "Societal Awareness" without expanding further and exploring complexities within these themes. More detailed analysis and varied participant quotes is required. 17. Do the results adequately reflect the diverse experiences of people with Long Covid? It seems the manuscript has focused on a limited range of experiences, potentially overlooking varied demographic and symptomatic representations. Consider broadening the range of experiences presented, ensuring a diverse representation of Long Covid impacts. 18. There might be repetition of ideas or concepts, with similar points made in multiple sections. The data presentation within themes is uneven, leading to a lack of clarity. Inconsistencies in how data is presented across different themes. Standardize the presentation of data across all themes for clarity and coherence. 19. Some parts of the manuscript's structure are unclear or disjointed, jumping between topics without clear transitions. Improve the structure with clear headings and logical flow of content. 20. The discussion may not effectively connect the study's findings with existing research, so the author/s need to improve this chapter by integrating the study's results more thoroughly with the current body of literature to demonstrate their significance and relevance. 21. Provide a more critical analysis of how the study's findings align with or differ from existing research, exploring potential reasons for these similarities or differences. 22. Expand the discussion on the implications of the study for practice, policy, and future research, providing a comprehensive view of its potential impact. 23. The conclusion does not strongly tie back to the initial research objectives. Clearly connect the conclusion to the original research objectives, demonstrating how the study has addressed these goals. 24. The concluding remarks might not effectively summarize the key findings and their importance. Ensure the discussion acknowledges the study's limitations and avoids overgeneralizing the results. 25. The concluding section might be underdeveloped. The conclusion may not fully capture the study's impact and implications. The conclusion primarily reiterates findings without delving into their broader significance. Expand on how the study contributes to existing knowledge, and what this means for future research, policy, and practice. 26. The conclusion briefly mentions limitations but does not explore their impact on the findings. Provide a more detailed discussion of the limitations and how they affect the interpretation and application of the results. 27. The conclusion suggests the need for policy changes but lacks specificity in recommendations. The author/s could provide more specific, actionable recommendations for policy, practice, or further research based on the study's findings.
--	---

VERSION 1 – AUTHOR RESPONSE

Reviewer 1
 ABSTRACT

MR #1: I don't think we should be adopting any new acronyms in the academic literature that start with people. It takes away from the person first language that the authors are trying to accomplish. I would suggest writing it out throughout the manuscript.

MR #1: We acknowledge this point by removing all cases of the acronym pwLC (people with Long Covid) from the revised manuscript.

MR #2: Did the authors collect demographic information on the participants besides gender? If so it should be in the participants section.

MR #2: The participant section in the abstract also includes the age range of participants. In the revised manuscript we have added the average disease duration. The participation section in the Abstract now reads as follows:

"Participants. 25 adults with Long Covid aged 19-76 years including 17 males and 8 females. Average disease duration was 80.1 weeks."

INTRODUCTION

MR #3: Line 43-54, where it says "participant-centred approach" this should be "patient centred approach" and there should be a couple examples of other types of illnesses that adopt this approach such as hepatitis c and opioid use disorder. <https://www.cmaj.ca/content/191/17/E460.short>

MR #3: The term "participant-centred approach" has been replaced with "patient centred approach". In the Introduction, we have referenced publications which have used and promote a patient centred approach in Long Covid research (7, 12). We do not feel that a description of the use of a patient centred approach in other illnesses would be relevant to the Introduction of this study.

METHODS

MR #4: Page 6, line 14-18: Were any patients asked to co-author the manuscript? Given them the proposer recognition for their involvement in the submitted work.

MR #4: Participants were invited to provide feedback but were not asked to co-author the manuscript.

RESULTS

MR #5: Page 9, line 5: Should this say "infectious disease" specialist, instead of "specialist" ...just saying specialist is extremely broad.

MR #5: The term "specialist" has been replaced with "infectious disease specialist" on Page 9 of the revised document.

MR #6: Page 9, line 35-45: Under the community support heading, would peer support be more appropriate? It seems they want to connect with other people with long COVID, that's what really stands out to me.

MR #6: We agree with this suggestion and have altered the "Community Support" subtheme to "Peer Support". Consistent with this change we feel that the description of community support regarding household help is more appropriate under the "Disability Support" subtheme, which is now reflected on page 10 of the revised document.

MR #7: There are a lot of oxford comma's missing through the manuscript, I think the authors should carefully review it and update this oversight.

MR #7: The grammar has been carefully reviewed and edited the grammar with commas added where necessary in the revised document.

Major Revision #1: There is no mention of vaccines throughout the manuscript...do we know if participants have been vaccinated or were asked any questions about vaccinations? I think there

should be some mention of vaccinations in the introduction and discussion section. If there weren't any questions asked to the participants, then maybe the authors could explain their rationale and include it in the limitation section.

Major Revision #1: The aim of this study was to qualitatively explore participant's lived experience and perceived needs associated with LC. Therefore, we consider exploration of patient medical history, such as vaccination status, as beyond the scope of the study. However, in the revised document we have added this point in the 'Strengths and Limitations' section of the Discussion, as follows:

"This study did not explore the impact of participant's medical history, such as COVID-19 vaccination status, on the results and this could be considered as a potential limitation. However, the focus of the study was to explore commonly described experiences and needs in Long Covid, rather than to analyse the diversity of individual factors or medical history impacting perceived needs."

Major Revision #2: There is no mention of any ethics board approval for this study...is this an oversight or was there no obtained?

Major Revision #2: Ethical approval is reported on page 5 of the original manuscript: "The study was approved by the University of Leeds Psychology Ethics Committee on 3rd January 2023 (PSC710)." This comment may be referring to NHS Ethical Approval which was not required for this study as we recruited a community-based sample rather than a patient sample.

Reviewer 2

1. The title could be more specific about the study's scope or the aspects of Long Covid being addressed. The author/s could refine the title to specify the key focus of the study, such as "Exploring Support and Healthcare Needs of People with Long Covid in the UK."

1. The authors disagree with the suggested change to the title and have chosen to retain the original title, "Identifying the Needs of People with Long Covid: A Qualitative Study in the UK". The original title summarises the key components of the study, the overall research objective to identify the needs of people with Long Covid and distinguishes this as a qualitative study. The reviewer's suggestion for "support and healthcare needs" to be included in the title narrows down the scope of the findings and could be misleading as healthcare needs were not a primary focus of this study.

2. The study's exclusion of the impact of additional long-term health conditions on participants' needs might limit the comprehensiveness of the findings. This omission could be critical, especially given the multifaceted nature of Long Covid and its intersection with other health issues.

2. The study did not assess the impact of additional long-term health conditions and this has been recognised as a potential limitation of the study in the original document in the 'Strengths and Limitations of this study' section on page 2 and in the Discussion on page 14, as follows: "This study did not consider how additional long-term health conditions may have impacted participant's perceived needs." Page 2.

"We did not consider how additional long-term health conditions may have contributed to participant's perceived needs, therefore it may not be possible to distinguish which needs are driven by LC or other co-morbidities." Page 15.

Whilst this may be considered as a potential limitation of the study, we did not seek to examine the differing needs between individuals based on medical history or illness comorbidity. The results and conclusions of the focus group discussions reflect common patterns of needs expressed by participants on a group-level rather than individual experiences.

3. On page 4, not specifying what "UK" stands for in the manuscript could be considered an oversight, particularly for an international audience. While "UK" commonly refers to the United Kingdom, it is a best practice in academic writing to avoid assumptions about readers' familiarity with abbreviations. The author/s should initially define this abbreviation, like stating "United Kingdom (UK)," before using the abbreviated form throughout the manuscript.

3. On page 4, "UK" has been corrected to "United Kingdom (UK)".

4. The introduction may lack clarity or focus on key research gaps in Long Covid research. The introduction broadly discusses Long Covid impacts without honing in on specific gaps the study aims to address.

6. The justification for the study by the author/s may not be clearly articulated. The rationale for conducting this study is not explicitly stated in the introduction.

4, 6. We agree that the justification for the study and gaps in the literature could be stated more explicitly in the Introduction. We have provided a stronger rationale in the revised document by clearly outlining the key gaps in the literature the study aims to address (changes underlined):

"Three years following the first cases of LC (11), reports continue to show lower life satisfaction, reduced happiness and greater symptoms of anxiety and depression in those with LC than those who have never been infected with the SARS-CoV-2 virus (3). Despite years of research, it remains unclear whether people with LC feel adequately supported with their condition in 2023. Previous qualitative studies exploring the lived experience of Long Covid have primarily focused on the difficulties of accessing healthcare and suggested quality improvements for healthcare services (4, 7, 8). Yet, these studies have also revealed the widespread issues faced by people with LC including the impact on employment, relationships, stigmatisation and mental health, without identifying actionable strategies to tackle these issues. Therefore, there is an urgent need to comprehensively examine the broad range of needs of people with LC to address these issues and identify strategies to support symptom management, recovery and quality of life."

5. There is limited context on the current state of Long Covid research and patient experiences provided by the author/s. General statements about Long Covid without detailed background on existing research or patient experiences.

5. In the original manuscript we attempted to provide a concise overview of Long Covid research relevant to this study, rather than providing a general literature review of Long Covid research. We acknowledge that the overview of patient experiences found in previous studies is described briefly and have included further detail as follows (changes underlined):

"While the immediate focus of LC research has been on reporting the symptoms, prevalence and risk factors, a few qualitative studies have reported the lived experiences of people with LC (4). The results reveal a debilitating condition which severely disrupts daily life due to the episodic and turbulent nature of symptoms, post-exertional symptom exacerbation and reduced physical and functional abilities (5-6). Many people with LC experience stigmatisation in society, apathy and a lack of understanding from friends and family due to the 'invisible' nature of the condition, with negative impacts on relationships and wellbeing (4, 5). Furthermore, the impact on self-identity is commonly reported, whereby people with Long Covid describe the difficulty of coming to terms with their new identity as an 'ill person' or their loss of professional identity where the condition impacted their employment (4, 5). The lack of access to services and limited treatment options contributed to feelings of hopelessness for recovery, poor mental health and wellbeing (5). Such qualitative studies have revealed the challenging experiences faced by people with LC in accessing UK healthcare services in the first 6 months of the COVID-19 outbreak, including difficulties achieving a diagnosis, navigating and accessing services and being taken seriously by health professionals (7-8)."

However, it should be acknowledged that the description of patient experiences in the Introduction is reflective of the lack of qualitative research exploring the experiences of Long Covid. To our knowledge, there are three main qualitative studies (references 5, 7, 8) and one systematic review (reference 4) that are relevant to this study. Overall, we aimed to ensure the Introduction section was concise and specific to this study.

7. Possible misalignment between study objectives and introduction content. The introduction does not directly lead to the specific objectives of the study. The author/s need to ensure that the

introduction seamlessly leads into the specific objectives of the study, making a direct connection between the background provided and the study's goals.

7. We feel this comment has been addressed in the revised document with the addition of the changes made in comments 5 and 6. The content in the Introduction now has a stronger emphasis on patient experiences described in previous research. Furthermore, the justification for the study is clearly stated, providing a stronger connection between the background to Long Covid research, gaps in the qualitative literature and the objective of the study to address the needs of people with Long Covid.

8. The choice of qualitative method is stated without detailed justification. Provide a rationale for why the qualitative method was chosen over other potential methods.

8. We have justified the choice of using qualitative methods and specifically the Framework Analysis in the Introduction in the revised manuscript:

“By utilising qualitative methods in this study, we aimed to explore the multifaceted experiences and perceived needs of individuals with Long Covid from a patient-centred perspective.... We aimed to conduct a qualitative exploration using the Framework Analysis (13) as this method is rigorous and systematic, providing a structured framework for analysing qualitative data while allowing for flexibility and reflexivity in the analysis process. The Framework Analysis was considered to be appropriate for this study as it facilitates the synthesis of commonly described themes or concepts within participant’s accounts, rather than deeply analysing and interpreting narratives as with other qualitative methods such as Narrative Analysis or Interpretive Phenomenological Analysis.”

9. Limited details on the criteria for participant selection. "Participants were eligible if they had self-reported LC symptoms and lived in the UK".

9. We appreciate that this is a brief description of the participant selection however, it accurately describes the approach taken. We aimed to take a broad, inclusive approach to recruitment to reduce barriers to participation and encourage people with LC across the UK to participate.

10. Further elaborate on how participants were selected, including any exclusion criteria. "Eight exploratory focus groups... were conducted in person and online". Provide more detailed descriptions of the data collection process, including how discussions were guided.

10. There were no exclusion criteria for participation in the study to ensure the study was inclusive for all people with Long Covid. Please refer to the Supplementary Information where the guided discussion questions are listed. In the revised manuscript, we have included further details about the data collection process, specifically about how the discussions were guided. This reads as follows (changes underlined):

“Each focus group lasted approximately 2 hours and was audio-recorded. Facilitators informally followed guidance questions (see Supplementary Information) intended to encourage discussion around their main concerns, symptoms and ideas for support and interventions. Discussions were self-guided by participants and facilitators were advised not to influence participant answers or direct the conversation. Guidance questions were asked as each discussion topic came to an end. This allowed for organic discussions to take place, led by the participants.”

11. Discuss ethical considerations in more depth, including how participant confidentiality and consent were handled.

11. On page 4 in the revised manuscript, we have included further detail about the steps taken to ensure participant confidentiality: “Confidentiality was ensured by removing personal identifiers from the data and by limiting access to only those involved in the data analysis”. In the original manuscript, participant consent is detailed as follows: “All participants provided written informed consent”.

12. Unclear representation of diverse experiences in participant selection. Please clearly describe efforts to include diverse demographics and acknowledge any limitations in representation.

12. We have further detailed our efforts for inclusivity of diverse demographics in the Methods section on page 4 of the revised document: "Participants were recruited via advertisement in the Leeds Community Healthcare LC services, the Chronic Pain & Fatigue Network at the University of Leeds (UoL), the English National Opera Breathe Programme and via social media in attempts to increase visibility to a range of networks... Online discussions were held to enhance the accessibility of the study ensuring those living outside the region or those with impaired mobility could attend."

We have included a description of the limitations in representation in the Discussion on Page 15: "Despite our efforts to advertise the study across the UK, most participants were located in West Yorkshire and therefore the results may not reflect those of people with LC across the UK. Furthermore, participants of this study were self-selecting therefore the sample may only reflect those who felt they had the mental capacity, time and opportunity to take part."

13. Vague explanation of data analysis steps: "The analysis followed the procedural steps for the Framework Method". It would be great if the author/s could provide a clearer trail of each step of the data analysis process, especially leading to how themes were derived. What about data saturation? Kindly elaborate how and when data saturation was reached and how this was determined during the analysis.

13. We agree that data analysis steps have been detailed too briefly and have edited this section of the Methods in the revised manuscript to clearly describe the procedural steps and outline when data saturation was reached. On page 5, the data analysis steps are now described as follows: "The data analysis followed the procedural steps for the Framework Method (13) in four stages: (a) data familiarisation was conducted by reading and rereading the transcripts and listening to audio recordings from which initial codes were identified; (b) a framework for coding the data was developed and agreed upon by each author; (c) the data was indexed through manually coding according to the framework in NVivo (R14.23.1, QSR International). To increase the reliability and validity of the data analysis, 25% of the data (2/8 transcripts) were independently double-coded by two analysts (A.M. and S.M.) and coding was compared. Any discrepancies were solved by consensus and the framework was altered accordingly. A.M. then applied the framework to all remaining transcripts to code the data. 20% of the coded data was randomly checked by the second data analyst (S.M.) in order to ensure consistency in the coding and reduce possible bias. Themes were generated through mapping and interpretation by summarising common concepts within the coded data, in relation to the research question (d). Data saturation was determined when no new themes were emerging from the analysis. To ensure the credibility of the data and analysis, the results were validated by discussing the themes within the research team and by seeking feedback from the participants. The research team involved in this study and manuscript refinement was interdisciplinary and interprofessional, combining expertise from Long Covid Rehabilitation, Neurology, Cognitive Psychology and Qualitative Methods."

14. Measures to ensure reliability and validity are not explicitly discussed. Discuss the steps taken to ensure the reliability and validity of the data collected and analyzed.

14. We acknowledge this point by including further detail in the Methods section on page 5 of the revised manuscript to detail the measures taken to ensure the reliability and validity of the data collected and analysed:

"To increase the reliability and validity of the data analysis, 25% of the data (2/8 transcripts) were independently double-coded by two analysts (A.M. and S.M.) and coding was compared. Any discrepancies were solved by consensus and the framework was altered accordingly... 20% of the coded data was randomly checked by the second data analyst (S.M.) in order to ensure consistency in the coding and reduce possible bias... To ensure the credibility of the data and analysis, the results were validated by discussing the themes within the research team and by seeking feedback from the participants."

15. The results chapter may lack specific details or examples with insufficient use of direct quotes

from participants to support the themes. Vague descriptions of participant experiences. Themes are stated with minimal direct quotations to substantiate them. The author/s must integrate more direct quotes from participants to illustrate points, provide authenticity and depth to the themes.

15. In the revised manuscript, we have included more direct quotes from participants in the Results section to illustrate the points made. The description of participant experiences is clearer with more specific details on the common patterns of experiences. This is particularly apparent in the "Clinical Support", "Employment Support", "Disability Support", "Peer Support" and "Educate Society" subthemes in which there is now a deeper analysis of participant's experiences and their needs.

16. Superficial theme development and analysis of participant responses. The author/s need to provide deeper analysis and critical insights into the data. For instance, the manuscript states themes such as "Support Systems" and "Societal Awareness" without expanding further and exploring complexities within these themes. More detailed analysis and varied participant quotes is required.

16. We have addressed these points in the revised document by re-evaluating the data analysis and providing more detailed accounts of themes with deeper analysis of participant responses. We have included theme summaries under "Support Systems" and "Societal Awareness" with more enhanced descriptions exploring participant experiences within subthemes. However, it should be noted that the focus of the study was on participant's perceived needs rather than an in depth analysis of experiences. Participant experiences have been explored to provide a context to each theme/need identified. We do not feel that it is appropriate for this study to deeply analyse experiences when this has been previously reported in the literature and was not an aim of this study.

17. Do the results adequately reflect the diverse experiences of people with Long Covid? It seems the manuscript has focused on a limited range of experiences, potentially overlooking varied demographic and symptomatic representations. Consider broadening the range of experiences presented, ensuring a diverse representation of Long Covid impacts.

17. The aim of the study was to assess the broad range of perceived needs, generalised across people with Long Covid. Therefore, the results reflect a group-level qualitative analysis of common experiences and needs identified by participants. We did not aim to explore an in-depth analysis of the diversity of individual needs, as this is beyond the scope of this study. Furthermore, the authors' and advisors' collective understanding is that the themes should be developed with an aim of answering the research question which we feel has been achieved. It should be noted that participants were invited to review the manuscript prior to submission and feedback showed that they agreed with the themes and confirmed that the results and conclusions represented their views. With regard to the diversity of demographic representations, the study employed an inclusive approach to recruitment to maximise participant diversity with no limitations to age, gender, symptom severity or duration. Symptomatic representations are addressed in a 2nd paper about this study and submitted to BMJ Open for consideration. Exploration of the differing needs between demographic factors (i.e. age, gender) or between symptoms was not an aim of this study.

18. There might be repetition of ideas or concepts, with similar points made in multiple sections. The data presentation within themes is uneven, leading to a lack of clarity. Inconsistencies in how data is presented across different themes. Standardize the presentation of data across all themes for clarity and coherence.

18. The reviewer is unclear about where there might be repetition of ideas and whether this was identified in the Results section or another section. We feel there may be repetition across the subthemes 'Educate Society' and 'Promote Awareness and Empathy' and have integrated this into one theme: 'Educate Society'. Other than this, we feel the themes presented are distinct and reflect the multifaceted needs of people with Long Covid across needs for support from government, the community and peers as well as needs for research and societal awareness. It should be acknowledged that the themes are not isolated concepts as all themes identify the needs of people

with Long Covid, therefore links and similarities between themes are to be expected. For example, the subthemes “Employment Support” and “Disability Support” are linked as some participants faced unemployment following LC and then required government financial assistance. However, these themes are distinct as “Employment Support” details the challenges people with LC face in the workplace whereas “Disability Support” addresses participants’ needs for government issued disability ‘aids’ including, financial assistance, the Blue Badge and household help.

The reviewer does not specify how the data presentation is uneven across themes and whether this refers to the format of the data presented or the quantity of data presented in each theme. We have acknowledged this point by standardising the format of quotes in the revised document. All quotes are centred, italicised and presented following a description of each theme in the revised document. The length and depth of each theme may be unbalanced, however this is reflective of the importance of this theme to participants. In other words, needs that were discussed more frequently and in greater depth by participants are described in more detail in the Results section.

19. Some parts of the manuscript’s structure are unclear or disjointed, jumping between topics without clear transitions. Improve the structure with clear headings and logical flow of content.

19. In the revised document, the structure of the Discussion has been improved by using subheadings including ‘Comparison findings with existing research’, ‘Implications for policy change’, ‘Strengths and Limitations’ and ‘Conclusions’. There has also been a focus on linking points clearly between paragraphs in the Discussion.

20. The discussion may not effectively connect the study’s findings with existing research, so the author/s need to improve this chapter by integrating the study’s results more thoroughly with the current body of literature to demonstrate their significance and relevance.

21. Provide a more critical analysis of how the study’s findings align with or differ from existing research, exploring potential reasons for these similarities or differences.

20, 21. The reviewer has provided similar suggestions in comments 20 and 21. We acknowledge that the Discussion could more effectively connect the study’s findings with existing research and have updated this in the revised manuscript. It should be noted that there is limited qualitative research investigating the lived experiences of LC and no other studies directly investigating the needs of people with LC. Therefore, whilst we have connected the findings with previous research, the findings of this study are unique. We did not identify any differences between the findings of this study and previous studies. The description of the findings in relation existing research now reads as follows in the revised manuscript:

“Comparison of findings with existing research

There are a few qualitative studies which have explored the lived experience of LC with which we can compare our findings (4, 5, 7, 8). The findings of this study are consistent with previous qualitative research demonstrating the general lack of understanding of LC. This results in experiences of judgement and stigmatisation in society and a lack of belief or empathy from employers and GPs, further contributing to the sense of burden and isolation associated with LC (4, 5, 7, 8). Participants in this study offered many suggestions for educating society and promoting awareness of LC including news reports, TV programmes and art. The most-reaching approach to enhancing the credibility of the condition, identified by participants, was government recognition of LC as a disability. Furthermore, the barriers to accessing healthcare found in this study reflect those described in previous studies including long waiting periods for referral, a lack of continuity of care and the lack of understanding or treatment options for the condition (7, 8). Many of the suggestions provided by participants in this study echo the proposed quality principles for services proposed by Ladds et al. (7). Specifically, the need for greater access to healthcare, multidisciplinary rehabilitation services, clinical responsibility and further development of medical knowledge and therapies offered. Further to these points, we suggest improving access to information during referral periods such as signposting to online information and community support to ensure people with LC are not ‘left in the dark’ during this time.”

22. Expand the discussion on the implications of the study for practice, policy, and future research, providing a comprehensive view of its potential impact.

25. The concluding section might be underdeveloped. The conclusion may not fully capture the study's impact and implications. The conclusion primarily reiterates findings without delving into their broader significance. Expand on how the study contributes to existing knowledge, and what this means for future research, policy, and practice.

27. The conclusion suggests the need for policy changes but lacks specificity in recommendations. The author/s could provide more specific, actionable recommendations for policy, practice, or further research based on the study's findings.

22, 25, 27. The reviewer has offered almost identical suggestion in comments 22, 25 and 27 to specify actionable recommendations for policy, practise and further research. This has been addressed in the original document as the Discussion includes a detailed description of the study's potential impact on policy change, specifically the need to legally recognise LC as a disability and the implications of this. The discussion provides a strong evidence base for this specific actionable recommendation for policy change, rather than offering generalised statements on the wider impacts which the study does not address.

We acknowledge that the concluding remarks could further highlight the specific outcome that LC should be legally recognised as a disability. In the revised document, the conclusion highlights the recommendation for Long Covid to be legally recognised as a disability and the implications of this. On page 16 in the revised document, the Conclusion reads as follows:

"In light of our exploration into the perceived needs of individuals with LC, our findings highlight a critical imperative for policy reform to recognise LC as a disability. Our investigation has brought to the forefront the multifaceted challenges faced by people with LC, from navigating employment uncertainties to managing daily activities and social interactions. By recognising LC as a disability, policymakers can pave the way for the implementation of comprehensive support systems that are tailored to the needs of individuals living with LC. Central to this is the imperative for policy initiatives to adopt a patient-centred, collaborative approach that actively involves individuals with LC in the decision-making process. Only through such inclusive policies can the diverse and individual needs of people with LC be adequately addressed. By bridging our findings with actionable policy recommendations, we aim to initiate meaningful strides towards enhancing the quality of life of individuals living with LC. This study recommends future studies should continue exploring these experiences in order to inform the development of innovative treatments and interventions to support the health and wellbeing of people with LC."

23. The conclusion does not strongly tie back to the initial research objectives. Clearly connect the conclusion to the original research objectives, demonstrating how the study has addressed these goals.

23. In the Discussion in the revised document, we have more effectively outlined the conclusions in relation to the original research objective by clearly articulating participants perceived need for greater support and recognition of their condition. On page 14, the Discussion begins with the following paragraph:

"The purpose of this qualitative study was to investigate the perceived needs of people with LC. The findings revealed an overarching message that people with LC require greater support and awareness of their condition. Firstly, participants expressed a critical need for robust support systems, including tailored employment assistance, disability support and peer support through organised groups. Second, there was a strong call for research to explore the underlying mechanisms of LC symptoms, develop new treatments, and improve clinical testing. Finally, participants emphasized the importance of raising awareness and promoting empathy for LC within society through public education initiatives. These findings underscore the complex challenges faced by individuals with LC

and highlight the urgent need for targeted interventions, research initiatives, and societal awareness efforts to address their diverse needs and enhance their quality of life.”

24. The concluding remarks might not effectively summarize the key findings and their importance. Ensure the discussion acknowledges the study's limitations and avoids overgeneralizing the results.

24. We feel that the addition of the paragraph in the revised document quoted in point 23 effectively summarises the key findings and avoids generalisation. The study's limitations are acknowledged in the 'Strengths and Limitations' section of the Discussion. Additional limitations have been clarified in this section and are detailed in point 26 below.

26. The conclusion briefly mentions limitations but does not explore their impact on the findings. Provide a more detailed discussion of the limitations and how they affect the interpretation and application of the results.

26. We acknowledge this point and have included further details of the study's limitations to the 'Strengths and Limitations' section and described the potential impact of these. In the revised manuscript, this section reads as follows (changes underlined):

“Strengths of this study include the accessibility of the focus groups via a choice of in-person or online attendance and the use of a second analyst to quality assess the framework and analysis to reduce bias in the coding procedure. The researchers and facilitators in this study had no personal experience of LC symptoms facilitating a more open minded approach to the interpretation of participant's experiences. However, inevitably this meant that they lacked shared experiences which could add richness to the interpretation. The inclusion of multiple facilitators in discussion groups may have introduced different styles of questioning and focuses of conversation. Despite this, there were common patterns in the data that emerged across discussion groups. The current study could be limited by a biased sample which consisted of a 2:1 ratio of female to male participants and predominantly middle-aged participants (-50%). LC is more prevalent in females than males (21) which may reflect a true gender difference or a reporting bias. Furthermore, despite our efforts to advertise the study across the UK, most participants were located in West Yorkshire and therefore the results may not reflect those of people with LC across the UK. This study did not consider how additional long-term health conditions may have contributed to participant's perceived needs, therefore it may not be possible to distinguish which needs are driven by LC or other co-morbidities. This study did not explore the impact of participant's medical history, such as COVID-19 vaccination status, on the results and this could be considered as a potential limitation. However, the focus of the study was to explore commonly described experiences and needs in Long Covid, rather than to analyse the impact of the diversity of individual factors or medical history on perceived needs.”